# Mix and match: Patchwork domain evolution of the land plant-specific Ca$^{2+}$-permeable mechanosensitive channel MCA

**Kanae Nishii**[1,2]*, **Michael Möller**[1], **Hidetoshi Iida**[3]*

**1** Royal Botanic Garden Edinburgh, Edinburgh, Scotland, United Kingdom, **2** Kanagawa University, Yokohama-shi, Kanagawa, Japan, **3** Department of Biology, Tokyo Gakugei University, Tokyo, Japan

* kanaenishii@gmail.com (KN); iida@u-gakugei.ac.jp (HI)

**Data Availability Statement:** R codes used in this study are published in protocol.io. [dx.doi.org/10.17504/protocols.io.bkqwkvxe] Gene trees and their alignment are available from TreeBASE. [http://purl.org/phylo/treebase/phylows/study/TB2:S26880]

## Abstract

Multidomain proteins can have a complex evolutionary history that may involve *de novo* domain evolution, recruitment and / or recombination of existing domains and domain losses. Here, the domain evolution of the plant-specific Ca$^{2+}$-permeable mechanosensitive channel protein, MID1-COMPLEMENTING ACTIVITY (MCA), was investigated. MCA, a multidomain protein, possesses a Ca$^{2+}$-influx-MCA$^{func}$ domain and a PLAC8 domain. Profile Hidden Markov Models (HMMs) of domains were assessed in 25 viridiplantae proteomes. While PLAC8 was detected in plants, animals, and fungi, MCA$^{func}$ was found in streptophytes but not in chlorophytes. Full MCA proteins were only found in embryophytes. We identified the MCA$^{func}$ domain in all streptophytes including charophytes where it appeared in E3 ubiquitin ligase-like proteins. Our Maximum Likelihood (ML) analyses suggested that the MCA$^{func}$ domain evolved early in the history of streptophytes. The PLAC8 domain showed similarity to *Plant Cadmium Resistance* (*PCR*) genes, and the coupling of MCA$^{func}$ and PLAC8 seemed to represent a single evolutionary event. This combination is unique in MCA, and does not exist in other plant mechanosensitive channels. Within angiosperms, gene duplications increased the number of MCAs. Considering their role in mechanosensing in roots, MCA might be instrumental for the rise of land plants. This study provides a textbook example of *de novo* domain emergence, recombination, duplication, and losses, leading to the convergence of function of proteins in plants.

## Introduction

Proteins are essential components in any biological organism, including plants. Each protein can be assembled from smaller units, termed domains, and a protein can consist of a single or multiple domains [1]. There exist several databases for the repository of protein domains found in biological organisms [2]. Pfam, for example, currently has 19,179 entries ([3]; Pfam v.34.0, released March 2021). During organismal evolution, protein domains can combine but also evolve *de novo*. These *de novo* domains can be further combined with other *de novo* or existing domains to create novel proteins [1]. During plant evolution, it has been suggested

The seed alignment of the MCAfunc domain is available in the new March 2021 release, v. 34.0, of Pfam [http://pfam.xfam.org/family/PF19584#tabview=tab0], as well as in the S19 Appendix of this study.

**Funding:** This work was supported by the Japan Society for the Promotion of Science (JSPS) [KAKENHI Grant Number 25120708], Ministry of Education, Culture, Sports, Science & Technology of Japan, to HI. KN's stay at RBGE is financially supported by the Edinburgh Botanic Garden Sibbald Trust [2018#18], JSPS [JSPS KAKENHI Grant Number 18K06375], and the Sumitomo Foundation [170204].

**Competing interests:** We declare that we have no competing interests.

that at least 500 novel protein domains unique to this evolutionary lineage have emerged [4]. A search of *Arabidopsis thaliana* proteomes suggested that 75% of its proteins have domains registered in Pfam [5]. This indicates that there still exist a significant amount of unknown protein domains or domain combinations even in well studied plants, let alone plants in general. The combination of domains is perhaps a cost-effective way for organisms to create novel proteins [1], and in *A. thaliana*, at least 25% of proteins have multiple domains [5].

Integral membrane proteins that mediate ion fluxes in response to mechanical stresses, including touch, wind, water flow, osmotic pressure, gravity, and cell division- and cell expansion-generated forces, are called mechanosensitive channels. To date, five groups of mechanosensitive channels are found in plants [6]. One of them is a group of MID1-COMPLEMENTARY ACTIVITY (MCA) proteins, which are shown to function as $Ca^{2+}$-permeable mechanosensitive channels [7, 8]. The genes encoding MCAs are found exclusively in the plant kingdom [7, 9], whereas genes encoding other groups of mechanosensitive channels are found in prokaryotes and/or eukaryotes. Therefore, MCAs are unique in terms of molecular evolution and it is interesting to investigate when and where the *MCA* genes appeared during plant evolution.

In *A. thaliana*, two paralogous *MCA* genes, *AtMCA1* and *AtMCA2* have been isolated, and their functions examined in great detail. The AtMCA1 protein is involved in touch sensing at the root tip and a hypoosmotic shock-induced increase in the cytosolic free $Ca^{2+}$ concentration [7]. AtMCA2 was reported to participate in $Ca^{2+}$ uptake at the roots [10]. In addition, AtMCA1 and AtMCA2 respond to membrane stretch to generate cation currents when expressed in *Xenopus laevis* oocytes [8]. Furthermore, MCA channels appear to have common functions in plants, based on studies on *Oryza sativa* OsMCA1 [11–13], *Nicotiana tabacum* NtMCA1, NtMCA2 [14], *Zea mays* CNR13 [15], and *Streptocarpus MCA*-like gene (as *Saintpaulia* in [16]; see [17]).

MCAs are approximately 420 amino acid (aa) residues long multidomain proteins. They retain the provisionally advocated ARPK domain (Amino-terminal domain of Rice putative Protein Kinases; 1–143 aa) [7], overlapping with the EF hand-like region at the N-terminal region (136–180 aa) (InterPro: IPR002048), and well-curated PLAC8 domain (Pfam ID: PF04749) at the C-terminal region (S1 Appendix). A coiled-coil motif is located in the middle of the proteins. An approximately 170 aa region at the N-terminus, covering the ARPK and the EF hand-like domains, has $Ca^{2+}$ influx activity and is proposed to be a functional domain of MCAs [18]. In this study, we defined the N-terminal region as the MCA functional (MCA-func) domain.

In previous work, an MCA Neighbor-Joining tree was published that included only a limited number of plants, *i.e.* one moss, one lycophyte, one gymnosperm, and eight angiosperms. The unrooted tree showed that MCA proteins were mostly grouped following the tree of life (*e.g.* tolweb.org/tree/), except for *Picea sitchensis* (gymnosperm) and *Linum usitatissimum* (angiosperm) [9]. However, information from this tree is insufficient to elucidate the evolutionary history of the protein family or their domains. To better understand the origin and evolution of MCA proteins in plants, a more comprehensive study is required. Thus, in the present study, wide-ranging phylogenetic analyses of MCA proteins were carried out on 25 viridiplantae proteomes and full MCA proteins of 55 streptophyte species. Here, for ranks, we followed the definition by Leliaert et al. [19] and NCBI Taxonomy Browser (https://www.ncbi.nlm.nih.gov/guide/taxonomy/), where viridiplantae include green algae (chlorophytes) and streptophytes, streptophytes include charophytes and embryophytes, and embryophytes (also termed as "land plants") include bryophytes (Hornworts, Liverworts, Mosses), lycophytes, ferns, gymnosperms and angiosperms. Since MCA is a multidomain protein, we focused on the evolution, origin and fate of each domain (MCA-func and PLAC8) as well as the full MCA protein. Comprehensive domain searches were carried out against the viridiplantae proteomes

that included two chlorophytes and two charophytes. The study represents an example for the evolutionary dynamics of a multidomain protein in plants.

## Materials and methods

### Proteomes, genome, and transcriptomes used in this study

Twenty-five proteomes including species ranging from chlorophytes to angiosperms, were downloaded from Uniprot (https://www.uniprot.org/) and plaza (https://bioinformatics.psb.ugent.be/plaza/versions/gymno-plaza/) (S2 Appendix). Genomes / transcriptomes of 55 plant species were explored to find the full MCA genes (S3 Appendix; KEGG: [20]; Phytozome: https://phytozome.jgi.doe.gov/pz/portal.html#; OneKP: [21]; NCBI Genome: https://www.ncbi/nlm.nih.gov; Fernbase: https://www.fernbase.org; EnsemblPlants: https://plants.ensembl.org). Recently, systematic studies returned the genus *Physcomitrella* to the genus *Physcomitrium* [22], but we used the name *Physcomitrella* in this study for consistency with the registered names in the databases. The proteome completeness information, *i.e.* BUSCO completeness values (BUSCO-C) were available for most taxa on the Uniprot database. The BUSCO-C values of proteomes from plaza database (*Cycas micholitzii*, *Taxus baccata*) were newly obtained in this study using BUSCO v.4.0.6 [23], by comparisons against viridiplantae_odb10 lineage datasets.

### Building profile Hidden Markov Models (HMMs)

Three profile HMMs were used in this study: for the full MCA protein the model in PANTHER (http://www.pantherdb.org/), 'PROTEIN MID1-COMPLEMENTARY ACTIVITY 1 (MCA1): PTHR46604.SF3.pir.hmm' (422 aa) was used. A new profile HMM was created with *hmmbuild* in HMMER v.3.1b2 package (http://hmmer.org/), for the 1–167 aa region of the putative MCA$^{func}$ domain (MCA$^{func}$.hmm; 167 aa). MCA$^{func}$.hmm was registered in Pfam v.34.0 (PF19584). The profile HMM of PLAC8, was obtained from Pfam v.33.1 (PLAC8.hmm: PF04749; 91 aa). Logos of the profile HMMs were generated with Skylign (http://www.skylign.org) (S4 and S5 Appendices).

### Building domain matrices

Proteomes were interrogated for the presence of the MCA$^{func}$ and PLAC8 domains using *hmmsearch* (HMMER package), and the default setting (*E* value < 10.00). In these, proteins with the 'full E-value' < 0.001 and > 30 aa homologous regions were kept for further analyses. The domain sequences were aligned with MCA$^{func}$.hmm and PLAC8.hmm, using *hmmalign*, respectively. The alignments were manually checked and corrected in BioEdit v.7.2.5 [24]. They were further trimmed to remove hypervariable regions with BMGE v.1.12 [25] on the Galaxy server (https://galaxy.pasteur.fr/).

The proteome of *M. polymorpha* subsp. *ruderalis* (UP000077202) did not include proteins with both domains, but the closely related *M. polymorpha* did. The full MCA sequence was found in the genome database of *M. polymorpha* subsp. *ruderalis* (NCBI Genome GCA_001641455.1; Mp_v4; LVLJ01003617.1:83933–90898), and was highly homologous to that in *M. polymorpha* (Phytozome v.12.0: Mapoly0134s0009) (S6 Appendix). Thus, the translated amino acid sequences of the genome region (LVLJ01003617.1:83933–90898) was used as "A0A176VHI1_MARPO*" (S7 and S8 Appendices).

## Domain-based phylogenetic analyses

Maximum likelihood (ML) analyses were carried out with PhyML v.3.0 [26] on the ATGC server (www.atgc-montpellier.fr), with Smart Model Selection (SMS) [27]. Tree topology searches using SPR were carried out, and SH-like αLRT values obtained for branch support. ML rapid bootstrapping analyses of 2000 replicates were performed for additional clade support with RAxML v.8 [28], using models selected with ToPALi v.2 [29].

The Phyml trees were examined with Notung v.2.9 [30] for determining the root of the trees. The required species tree for this analysis (S9 Appendix) followed the Tree of Life Web Project (http://tolweb.org) and Angiosperm phylogeny website v.14 [31]. The bryophyte relationships followed [32]. For the MCA$^{func}$, the proteins in the charophyte *K. nitens* were suggested as root (S10 and S11 Appendices). For the PLAC8 domain tree, no strong root position

**Table 1. Number of proteins found in proteomes.** Result of profile HMM searches of MCA$^{func}$ and PLAC8 domains in proteomes of 25 taxa across viridiplantae. Number of proteins retaining the MCA$^{func}$ or PLAC8 domains ($E$ value < $10^{-3}$) are listed and arranged following the Tree of Life (see S2, S7, and S8 Appendices).

| Vernacular name | ID | Taxon | No of proteins with | |
|---|---|---|---|---|
| | | | MCA$^{func}$ | PLAC8 |
| Chlorophytes | CHLRE | *Chlamydomonas reinhardtii* | 0 | 12 |
| Chlorophytes | VOLCA | *Volvox carteri* f. *nagariensis* | 0 | 7 |
| Charophytes | KLENI | *Klebsormidium nitens* | 2 | 13 |
| Charophytes | CHABU | *Chara braunii* | 8 | 12 |
| Bryophytes | MARPO | *Marchantia polymorpha* | 9 | 13 |
| Bryophytes | MapoRu | *Marchantia polymorpha* subsp. *ruderalis* | 9 | 7 |
| Bryophytes | PHYPA | *Physcomitrella patens* | 9 | 23 |
| Lycophytes | SELML | *Selaginella moellendorffii* | 17 | 15 |
| Gymnosperm | CMI | *Cycas micholitzii* | 1 | 12 |
| Gymnosperm | TBA | *Taxus baccata* | 7 | 10 |
| Angiosperm | AMBTC | *Amborella trichopoda* | 7 | 9 |
| Angiosperm | MUSAM | *Musa acuminata* subsp. *malaccensis* | 9 | 26 |
| Angiosperm | ORYSJ | *Oryza sativa* subsp. *japonica* | 29 | 23 |
| Angiosperm | MAIZE | *Zea mays* | 11 | 25 |
| Angiosperm | SORBI | *Sorghum bicolor* | 13 | 19 |
| Angiosperm | AQUCA | *Aquilegia coerulea* | 10 | 10 |
| Angiosperm | VITVI | *Vitis vinifera* | 7 | 20 |
| Angiosperm | POPTR | *Populus trichocarpa* | 12 | 26 |
| Angiosperm | MEDTR | *Medicago truncatula* | 10 | 22 |
| Angiosperm | CUCSA | *Cucumis sativus* | 5 | 10 |
| Angiosperm | GOSRA | *Gossypium raimondii* | 14 | 29 |
| Angiosperm | BRAOL | *Brassica oleracea* var. *oleracea* | 7 | 29 |
| Angiosperm | ARATH | *Arabidopsis thaliana* | 5 | 20 |
| Angiosperm | ERYGU | *Erythranthe guttata* | 9 | 27 |
| Angiosperm | SOLLC | *Solanum lycopersicum* | 7 | 19 |
| | *sum* | | 217 | 438 |

The number of MCA$^{func}$ domain proteins within a proteome varied between species. In charophytes, *Klebsormidium nitens* had two proteins, but *Chara braunii* eight. In angiosperms, monocots possessed generally higher numbers between nine and 29, whereas dicots five to 14 per species. The number of PLAC8 domain proteins was between seven and 30 per species, the lowest in the liverwort *Marchantia polymorpha* subsp. *ruderalis* and the highest in *Brassica oleracea* var. *oleracea*. The more complete liverwort proteome of *Marchantia polymorpha* (BUSCO 96.7% in UniProt) had 14 PLAC8 genes. The low number in *M. polymorpha* subsp. *ruderalis* (BUSCO 91.3%) might be explained by the incompleteness of its proteome. The full MCA protein with both of MCA$^{func}$ and PLAC8 domains was not found in charophytes. In streptophytes, at least one and up to three full MCA proteins were found per species.

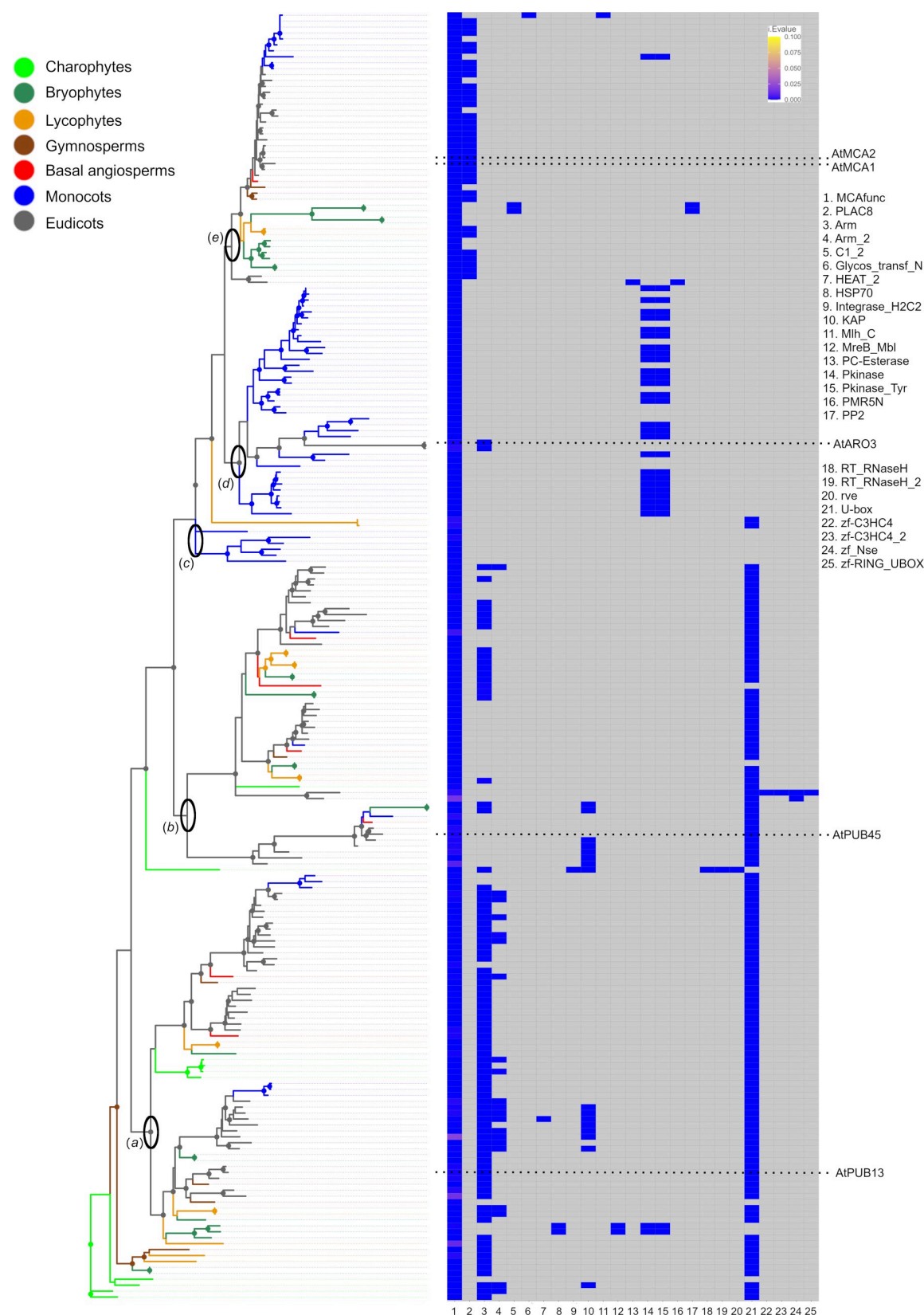

**Fig 1. ML tree of MCA^func domains and their partner domains predicted by HMMER website searches.** Left: ML tree. (*a*) Clade associated with E3 ubiquitin ligase AtPUB13. (*b*) Clade associated with AtPUB45. (*c*) MCA^func only proteins. (*d*) Clade associated with AtARO3 and monocot U-box containing protein kinase like proteins. (*e*) MCA clade. Clades supported with α-LRT SH-like values > 0.8 indicated with circles at the nodes. Right: Domain individual *E* values (i.Evalue) resulting from HMMER website searches are shown as a heatmap. Absence of domains indicated in grey. The original tree and heatmap are shown in S11 and S16 Appendices. Numbers in legends to the right indicate the domain names (x-axis of the heatmap).

was indicated (S12 Appendix), and thus a midpoint rooted tree was shown for ease of visualization (S13 Appendix).

## Partner domain HMMER searches of MCA^func and PLAC8 domains

The retained proteins possessed either MCA^func or PLAC8, or both domains. To determine the exact domain composition of these proteins, they were searched against the Pfam with *hmmscan* in HMMER (https://www.ebi.ac.uk/Tools/hmmer/). Based on their E-values they were visualized through R [33], as a colour-coded rooted phylogeny and heatmap utilizing ggplot2 [34], ggtree [35], ape v.5.0 [36], and phytools [37]. In some cases where two closely related domains were predicted for the same genome position, or domain duplications were involved, the domains with the lowest *E* values were selected. The data was also used for the schematic illustrations of representative domain structures visualized by R with a modified script based on Brennan (https://rforbiochemists.blogspot.com/2015/11/drawing-protein-domain-structure-using-r.html).

## Phylogeny analysis of full MCA protein sequences

BLAST searches were carried out on plant genome and transcriptome databases using the AtMCA1 protein sequence. The found sequences were further evaluated using *hmmsearch* with MCA^func.hmm and PLAC8.hmm. Only genes possessing both domains were included in the phylogenetic analysis. The positions of MCA sequences in genomes were examined where it was possible, and only one transcript sequence involved, *e.g.* in *Selaginella moellendorffii*, one MCA genome sequence found, whereas two identical proteins are present in the proteome (UP000001514). Thus, only one MCA from *S. moellendorffii* was included in the analyses. A phylogenetic tree was built with Phyml v.3.0, and subjected to Notung analyses for rooting. The bryophytes were suggested as likely root (S14 Appendix).

Because the study focussed on MCA, we specifically analysed gene duplication events for the full MCA protein tree in a Notung reconcile analysis (tree rearranged with Edge Weight Threshold = 0.6). The species tree used here (S15 Appendix) followed the Angiosperm phylogeny website v.14 [31]. The bryophyte relationship followed [32]. The relationships within angiosperm followed [38], for Brassicaceae [39], for Fabaceae [40], and for Poaceae [41].

## Results

### MCA^func domain found in streptophytes, MCA^func+PLAC8 in land plants

To determine the distribution of MCA proteins in viridiplantae, 25 proteomes (see Table 1) were interrogated for domains of the MCA protein, MCA^func and PLAC8, with profile HMMs using HMMER. In total, 217 proteins were found possessing only the MCA^func domain, 438 with only the PLAC8 domain, and 32 possessing both domains (Table 1; S7 and S8 Appendices). The MCA^func domain was only present in streptophytes, whereas the PLAC8 domain was found in all proteomes examined in this study (Table 1).

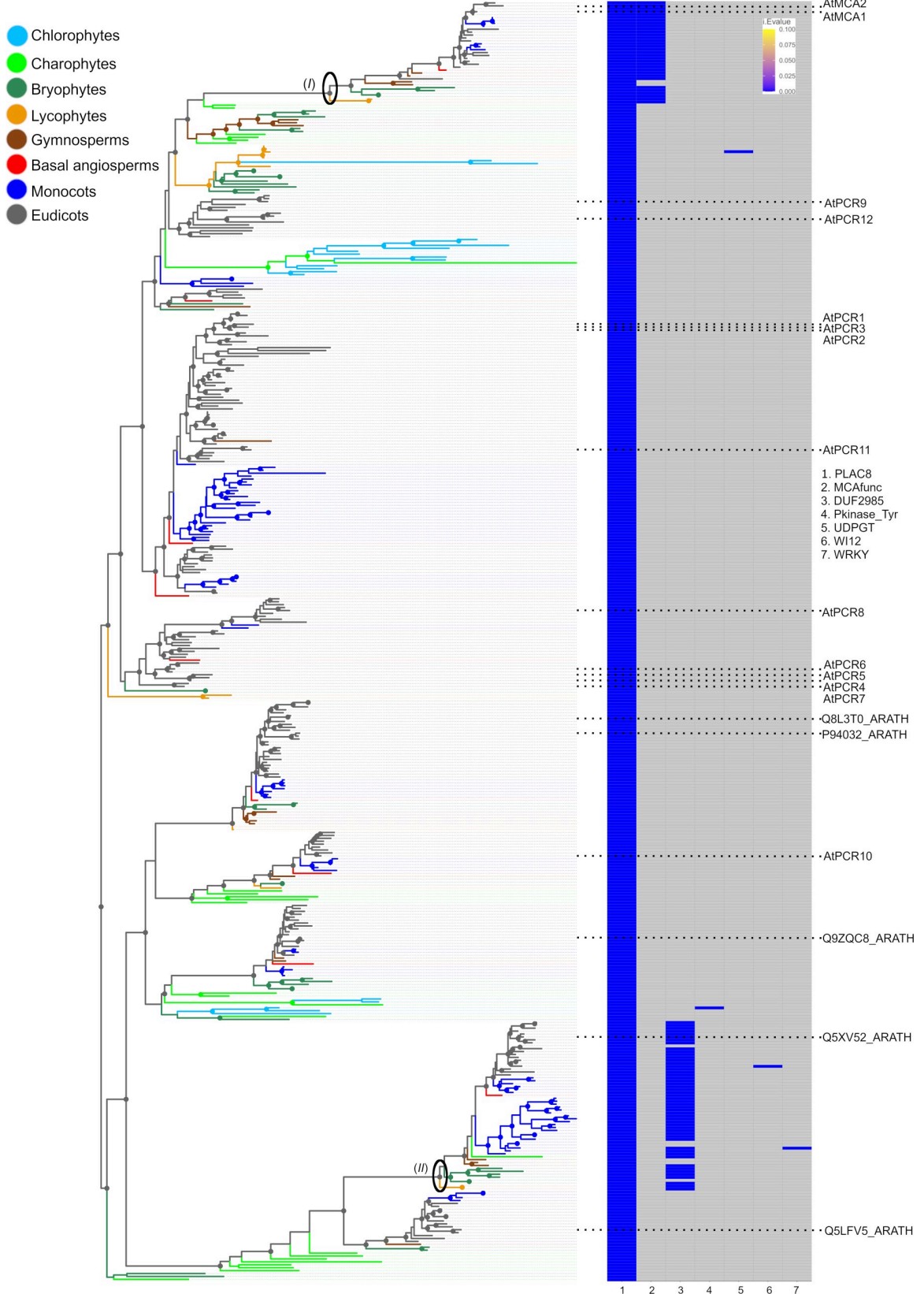

**Fig 2. PLAC8 domain tree and their partner domains.** Left: PLAC8 domain ML tree. (*I*) MCA 'Clade *I*'. (*II*) 'Clade *II*' with proteins retaining DUF2985 + PLAC8. Clades supported with α-LRT SH-like support values > 0.8 indicated with circles at the node. Right: Domain combinations observed in the PLAC8 domain containing proteins associated with the ML tree (left). Domain individual *E* values (i.Evalue) resulting from HMMER website searches are shown as a heatmap. Absence of domains indicated in grey. The original tree and heatmap are shown in S13 and S17 Appendices. Numbers in legend to the right indicate the domain names (x-axis of the heatmap).

## MCA^func and PLAC8 domain phylogenies

Since MCA is a multidomain protein, we studied the phylogenetic relationships of the domains MCA^func and PLAC8 separately. In the MCA^func domain Maximum Likelihood (ML) tree of 217 domain sequences, the samples included clustered according to the presence of partner domains (Fig 1; S11 and S16 Appendices). For example, samples of charophytes and 'Clade *a*', that included AtPUB13 (RING-type E3 ubiquitin ligase), had U-box (PF04564.15), Arm (PF00514.23) or Arm_2 (PF04826.13) as partner domains to MCA^func (Fig 1). Arm and Arm_2 are overlapping domains. 'Clade *a*' contained two major clades each including all streptophyte lineages, suggesting a gene duplication. 'Clade *b*' (Fig 1; S11 Appendix) also contained two main clades including most streptophyte lineages, suggesting a further duplication, where most proteins in one clade had lost the Arm domain. The following clades '*c*' and '*d*' contained mostly monocot-specific undescribed or potential protein kinase proteins (Fig 1) (*e. g.* rice Q2QZY3). 'Clade *e*', is the MCA protein clade including AtMCA1 and AtMCA2, where the majority of MCA^func domain proteins were partnered with the PLAC8 domain, which suggested that MCA as the derived proteins. A few proteins scattered across 'Clade *e*' had lost PLAC8 (Fig 1E), but there always was at least one protein with MCA^func plus PLAC8 present in each species (Table 1). Some MCA proteins had obtained an alternative partner domain such as C1_2 (PF03107.16) and PP2 (PF14299.6) (*M. polymorpha*), Pkinase (PF00069.25), or Pkinase-Tyr (PF07714.17) (*M. acuminata*) C1_2 and Mlh1_C (PF16413.5) (*Z. mays*).

In *Zea mays*, A0A1D6PNG8 and A0A1D6F850 hold the protein name "MCA1" in UniProt, but they in 'Clade *d*' and also lacked PLAC8 but retained Pkinase or Pkinase_Tyr. On the other hand, CNR13 and A0AD6JP06 were found to be proper MCAs since they were in the MCA clade ('Clade *e*') and possessed PLAC8 (S11 and S16 Appendices), as previously reported [15].

The phylogeny of the other MCA domain, PLAC8, was also examined phylogenetically. In the PLAC8 domain ML tree of 438 domain sequences, the samples also clustered according to their partner domains (Fig 2; S13 and S17 Appendices). Most of PLAC8 domain proteins appeared as single domain proteins, but the MCA clade ('Clade *I*') retained MCA^func, while another clade ('Clade *II*') retained a DUF2985 (PF11204) domain (Fig 2; S13 and S17 Appendices) with unknown functions. In *A. thaliana*, PLAC8 single domain proteins are registered as "Plant Cadmium Resistance proteins (PCR)", with the function to reduce cadmium uptake [42]. The MCA clade appeared to be closely related to a clade including AtPCR9 and AtPCR12. All proteins in Clade *I*, except for two, retained MCA^func and PLAC8. In gymnosperms, two proteins per species were found, with one having MCA^func while the other lacked it. The ML tree topology and distribution of partner domains suggested that the coupling between MCA^func and PLAC8 domains occurred once in the plant lineage, possibly in the common ancestor of embryophytes, and was sometimes lost after gene duplication events but was always retained in at least one copy.

## U-box and Arm are original partners of the MCA^func domain

While the PLAC8 domain commonly existed within the plant, animal, and fungi kingdoms, the MCA^func domain was only observed in streptophytes in the plant kingdom. Thus, the

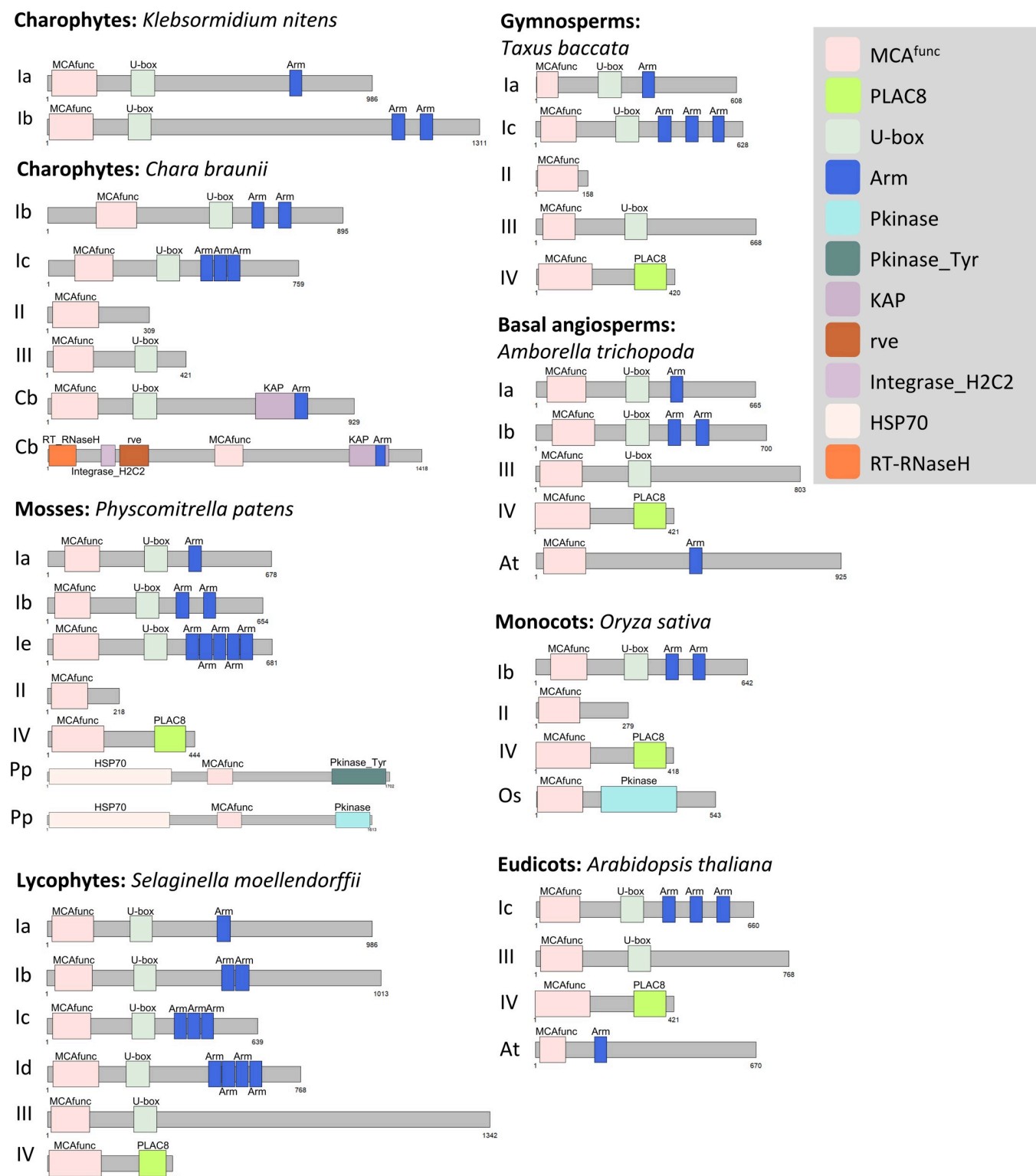

**Fig 3. Schematic illustration of protein domain combinations of the MCA$^{func}$ domain proteins in representative proteomes.** Type I, MCA$^{func}$ + U-box + Arm, number of Arm copies varies from 1 to 5 (Ia to Ie). Type II, MCA$^{func}$ only. Type III, MCA$^{func}$ + U-box. Type IV, MCA$^{func}$ + PLAC8. The well-studied MCA proteins in *A. thaliana*, AtMCA1 and AtMCA2, are type IV proteins. Lineage specific combinations are indicated by initials of the species, Cb, Pp, Os, At respectively.

**Table 2. Domain partners of MCA^func domain and their combinations found in proteomes across viridiplantae.**

| ID | Taxon | Domain combinations (Types) | | | | | | |
|---|---|---|---|---|---|---|---|---|
| | | **I** | **II** | **III** | **IV** | **Os** | **At** | *sum* |
| CHLRE | *Chlamydomonas reinhardtii* | 0 | 0 | 0 | 0 | 0 | 0 | 0 |
| VOLCA | *Volvox carteri* f. *nagariensis* | 0 | 0 | 0 | 0 | 0 | 0 | 0 |
| KLENI | *Klebsormidium nitens* | 2 | 0 | 0 | 0 | 0 | 0 | 2 |
| CHABU | *Chara braunii* | 4 | 1 | 1 | 0 | 0 | 0 | 6 |
| MARPO | *Marchantia polymorpha* | 4 | 1 | 1 | 1 | 0 | 0 | 7 |
| MapoRu | *Marchantia polymorpha* subsp. *ruderalis* | 4 | 2 | 0 | *1 | 0 | 0 | 7 |
| PHYPA | *Physcomitrella patens* | 3 | 2 | 0 | 2 | †2 | 0 | 9 |
| SELML | *Selaginella moellendorffii* | 12 | 0 | 3 | 2 | 0 | 0 | 17 |
| CMI | *Cycas micholitzii* | 0 | 0 | 0 | 1 | 0 | 0 | 1 |
| TBA | *Taxus baccata* | 3 | 2 | 1 | 1 | 0 | 0 | 7 |
| AMBTC | *Amborella trichopoda* | 3 | 0 | 3 | 1 | 0 | 1 | 8 |
| MUSAM | *Musa acuminata* subsp. *malaccensis* | 1 | 0 | 4 | 3 | 1 | 0 | 9 |
| ORYSJ | *Oryza sativa* subsp. *japonica* | 1 | 14 | 0 | 1 | 7 | 0 | 23 |
| MAIZE | *Zea mays* | 1 | 2 | 1 | 2 | 4 | 0 | 10 |
| SORBI | *Sorghum bicolor* | 1 | 2 | 0 | 1 | 9 | 0 | 13 |
| AQUCA | *Aquilegia coerulea* | 4 | 0 | 2 | 1 | 0 | 0 | 7 |
| VITVI | *Vitis vinifera* | 3 | 0 | 3 | 1 | 0 | 0 | 7 |
| POPTR | *Populus trichocarpa* | 6 | 0 | 2 | 2 | 0 | 0 | 10 |
| MEDTR | *Medicago truncatula* | 4 | 0 | 2 | 2 | 0 | 0 | 8 |
| CUCSA | *Cucumis sativus* | 2 | 1 | 0 | 1 | 0 | 0 | 4 |
| GOSRA | *Gossypium raimondii* | 10 | 0 | 2 | 1 | 0 | 0 | 13 |
| BRAOL | *Brassica oleracea* var. *oleracea* | 3 | 0 | 0 | 3 | 0 | 1 | 7 |
| ARATH | *Arabidopsis thaliana* | 1 | 0 | 1 | 2 | 0 | 1 | 5 |
| ERYGU | *Erythranthe guttata* | 3 | 1 | 3 | 2 | 0 | 0 | 9 |
| SOLLC | *Solanum lycopersicum* | 4 | 0 | 1 | 1 | 0 | 0 | 6 |
| *sum* | | 79 | 28 | 30 | 32 | 21 | 3 | |

The result of HMMER searches of MCAfunc domain partners and their combinations are listed and arranged following the Tree of Life. The types of domain combinations are described as follows: Type I: MCA^func + U-box + Arm/Arm_2, Type II: MCA^func only, Type III: MCA^func + U-box, Type IV: MCA^func + PLAC8 (MCA protein type), Os (monocot type): MCA^func + Pkinase/Pkinase_Tyr. † -with HSP70. At (ARO3 type): MCA^func + Arm
*—not found in the proteome but in the genome (see S3, S6–S8 Appendices).

MCA^func domain might be the key domain for the MCA protein, and we further assessed the coupling of the MCA^func domain with its partner domains. The predicted domain combination for *K. nitens* (charophyte) was MCA^func + U-box + Arm (type I) (Fig 3). Type I was found in all species, but the number of Arm domains varied from one to five. In *C. braunii* (charophyte), in addition to type I, MCA^func only (type II) and MCA^func + Ubox (type III) and lineage specific types (Fig 3Cb) were found. In *P. patens* (bryophyte-moss), type I, type II, the MCA type (MCA^func + PLAC8: type IV), and also lineage specific types were found (Fig 3Pp). The proteome of ferns was not available, but type I to type IV, and lineage specific combinations were widely observed from lycophytes to angiosperms. The *O. sativa* MCA^func + Pkinase (Os) type was widely present in angiosperm monocots, and in the moss *P. patens*, possessing an additional HSP70 domain (Pp). The *A. thaliana* MCA^func + Arm: ARO3 (At) type was only observed in the angiosperms *A. thaliana*, *Brassica oleracea*, and *Amborella trichopoda* (Fig 3; Table 2).

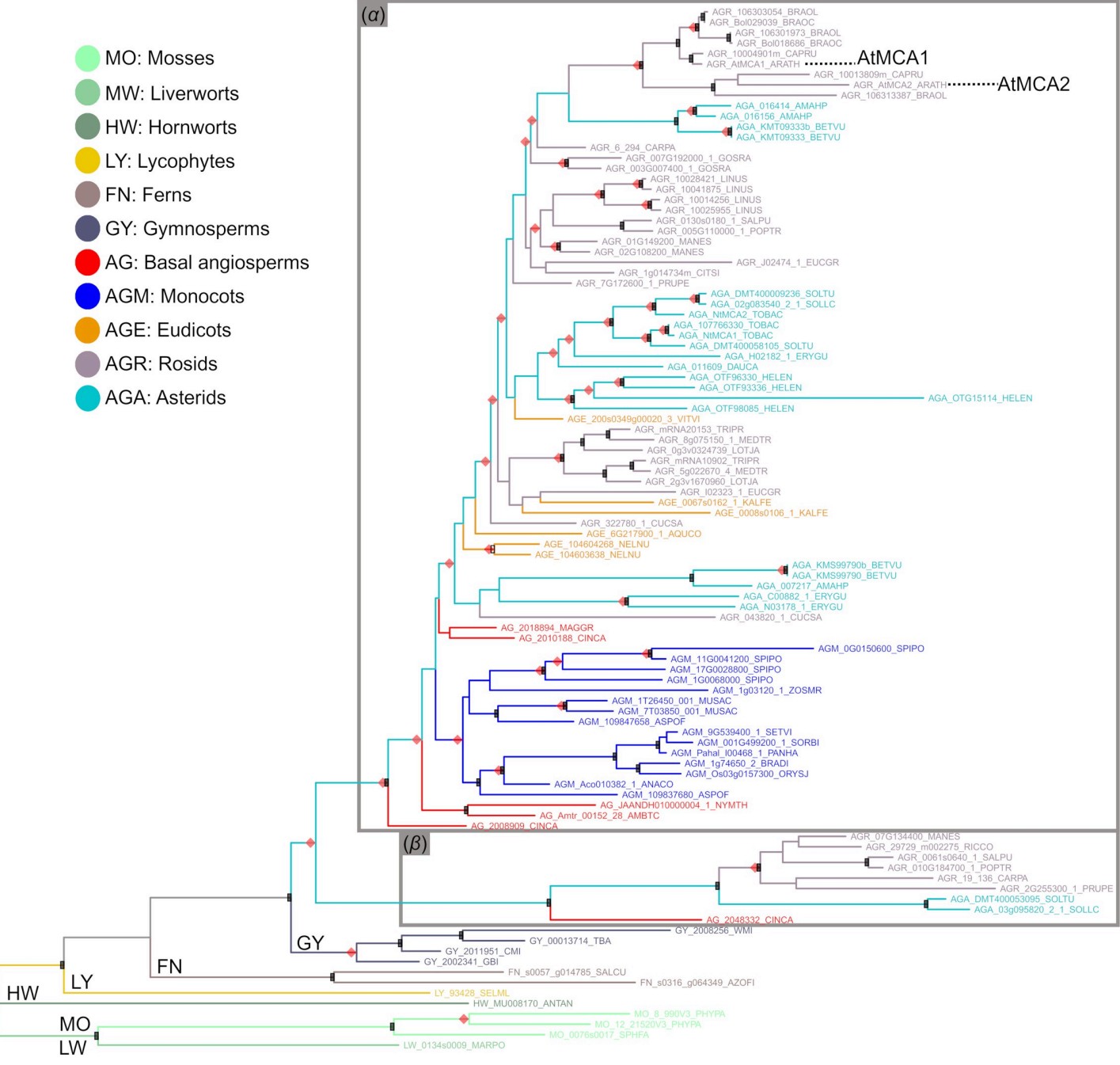

**Fig 4. MCA protein ML tree, rooted on the proteins of bryophytes.** Proteins have diverged into two clades in angiosperms ('α', 'β'). (α) Main MCA clade with all angiosperm species. (β) Diversified MCA clade showing variation at the N-terminus (see Fig 5). Black squares at the nodes indicate high values in both Phyml αLRT (> 0.7) and RaxML (> 70%) clade support analyses. Red diamonds at the nodes indicate Notung-inferred duplication events.

## Full MCA protein phylogeny, duplication and diversification in land plants

In order to unravel the history of MCA proteins in plants, a phylogeny of 106 full MCA proteins from 55 embryophyte species was reconstructed. The full MCA proteins include both MCA$^{func}$ and PLAC8 domain sequences. In this analysis, the basal grades in the ML tree, from bryophytes to gymnosperms followed the tree of life relationships (Fig 4). The angiosperm

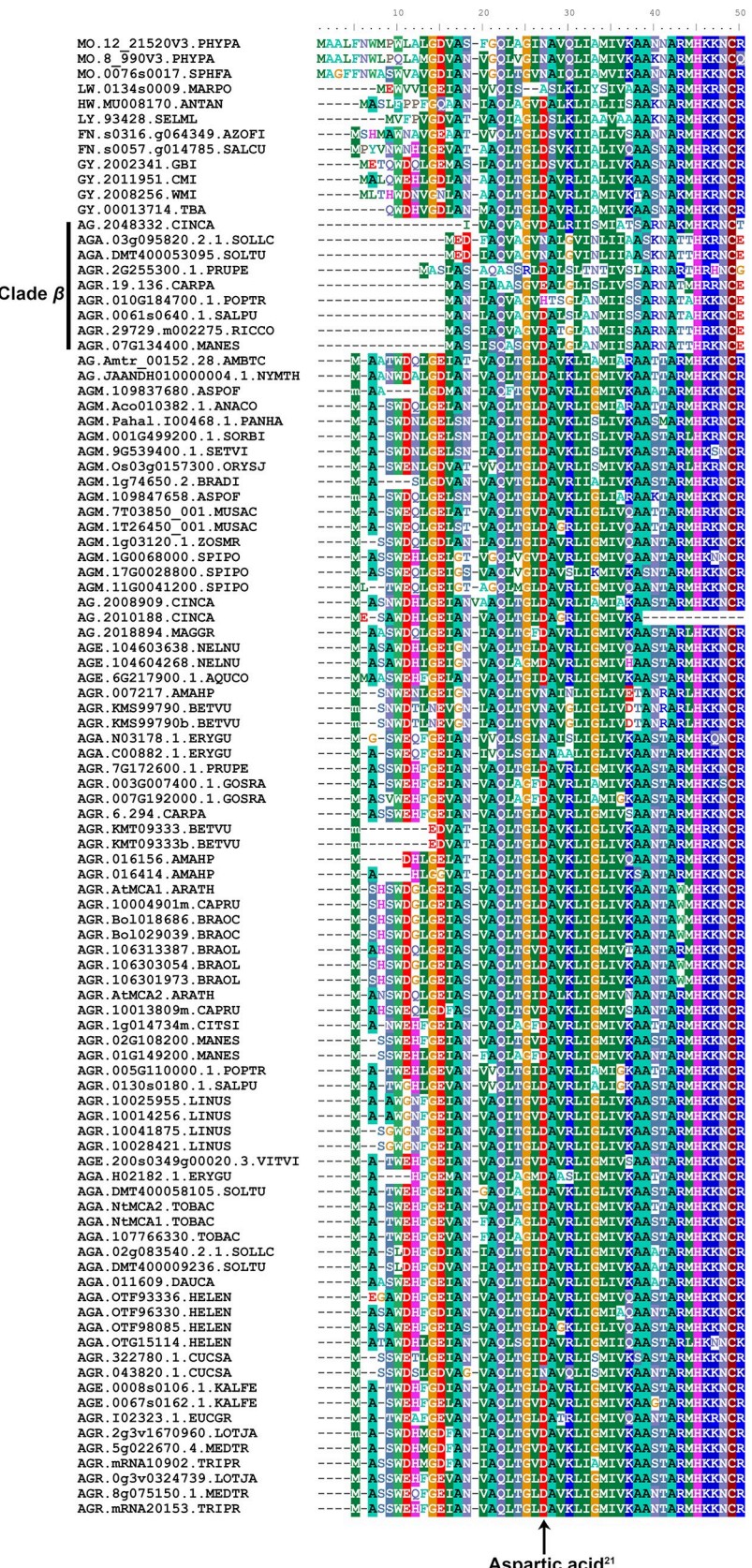

Clade β

Aspartic acid[21]

Fig 5. Conservation and variations in the N-terminus region of MCA proteins. The gene IDs are listed in S3 Appendix. The functional site of the 21st position of Aspartic acid (Aspartic acid21, arrow) in AtMCA1 ("AGR. AtMCA1.ARATH" in the Fig 5) was well conserved within the MCA proteins, except in mosses and liverworts. Variation in the N-terminus region was observed within angiosperm species, while at least one MCA protein kept the Aspartic acid21. The angiosperm clade β MCA proteins lack approximately 10 amino acids in the N-terminal region.

MCAs formed two clades. The majority of proteins fell in 'Clade α' including proteins of all angiosperm species analysed in this study. Only nine proteins formed 'Clade β', representing the orders Laureales (*Cinnamomum camphora*), Malpighiales, Rosales, Solanales and Brassicales (*Carica papaya*). These showed an MCA diversification and lacked approximately 10 aa in the N-terminal region (Fig 5). The predicted functional site of AtMCAs, the 21st position of aspartic acid ($Asp^{21}$; Fig 5, arrow) [18], was different in mosses (asparagine) and liverworts (alanine). Hornworts, on the other hand, retained $Asp^{21}$. At least one MCA per species retained $Asp^{21}$ from lycophytes to angiosperms (Fig 5).

A maximum of 39 duplication events were estimated across the ML tree, with two outside angiosperms (Fig 4, S18 Appendix). Three duplication events were inferred prior or at the point of diversification of angiosperms. Within the angiosperms, duplications were scattered among the lineages, but the superrosid clade stood out with an accumulation of six inferred duplications events. For several species repeated duplication events were inferred, *e.g.* three in *Linum usitatissimum* and two in *Beta vulgaris* and *Helianthus annuus* (S18 Appendix).

## Discussion

The evolution of multidomain proteins can be complex, and may involve *de novo* domain evolution, recruitment of existing domains as partners, and recombination and domain losses [1]. In the present study where the evolution of the multidomain protein MCA was examined in detail, the results showed that it represents an example with a complex evolutionary history.

Our comprehensive proteome interrogation with profile HMMs suggested that the MCA-func domain [18], formerly subscribed as ARPK domain plus EF hand-like [7], is a well conserved domain among plants. Accordingly, MCA can be described as a multidomain protein composed of the MCAfunc and the PLAC8 domains. PLAC8 is widely observed in eukaryotes as seen in our profile HMM searches, in which we found it in all proteomes we examined (Table 1). The MCAfunc domain, on the other hand, was streptophyte-specific and not found in chlorophytes, suggesting that the domain originated in the common ancestor of streptophytes, *i.e.* charophytes plus embryophytes (Fig 6) [4].

The MCAfunc evolution further included domain recruitment, recombination and losses. The E3 ubiquitin ligase-type proteins (type I in Fig 3) found in charophytes, represent an ancestral combination (Table 1). Type I proteins were found in most streptophytes, except *Cycas micholitzii* possibly due to the incompleteness of its proteome. The ancestral charophyte *K. nitens* retained only the type I, while in the more derived *C. braunii*, MCAfunc obtained different partner domains or lost them all (Fig 3). Although domain-losses need to be seen with caution in some species included here due to their proteome incompleteness, single MCAfunc domain proteins were also observed in well-assembled genomes such as *P. patens* and *O. sativa*, supporting the existence of single-domain MCAfunc proteins (Fig 1 and Fig 3). Lineage-specific domain combinations were also observed in angiosperms, such as the Os- and At-types. Intriguingly, the At-type was only found in Brassicaceae and *A. trichopoda*, but perhaps due to unrelated parallel evolutionary events (Table 1).

A key event for the MCA evolution seemed to be the partnering of MCAfunc and PLAC8 first recruited in the common ancestor of embryophytes. MCA is seemingly streptophyte-specific and might play some basic roles, perhaps as a mechanosensor, for habit expansion to terra

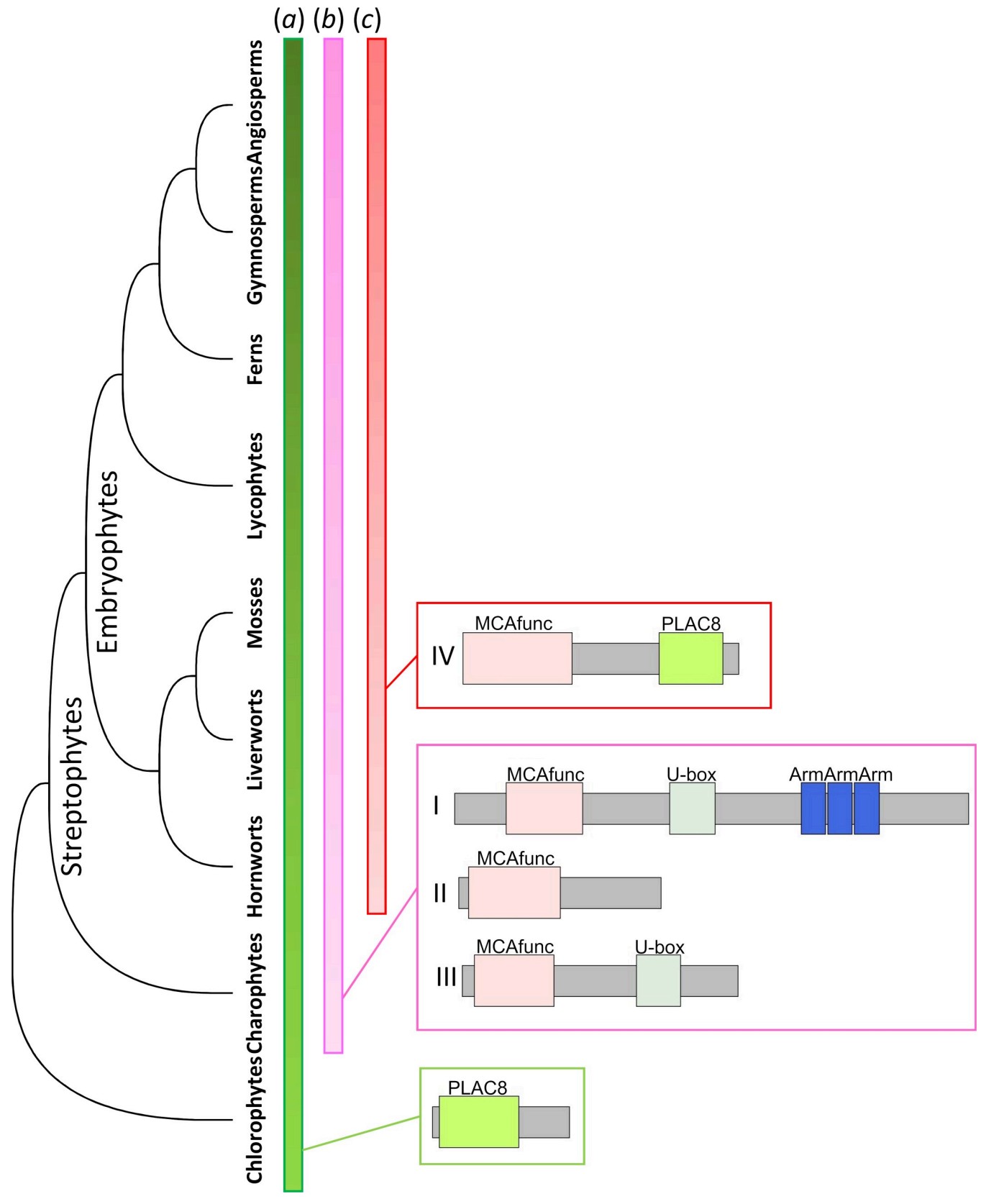

**Fig 6. Schematic illustration of the MCA protein evolution.** PLAC8 domain genes were observed from chlorophytes to angiosperms, and many were single domain proteins (*a*). In the common ancestor of streptophytes, type I, type II, and type III domain combinations evolved (*b*). In the common ancestor of land plants, *i.e.* embryophytes, type IV domain combinations (MCA type, *i.e.* MCA^func + PLAC8) evolved (*c*). In addition to the commonly observed types I-IV, lineage specific combinations were also observed.

firma [9]. A previous study indicated that the Asp$^{21}$ of MCA^func domain is crucial for Ca$^{2+}$ uptake [18]. Since Asp$^{21}$ is diversified in mosses and liverworts, it could be hypothesized that their MCAs do not have Ca$^{2+}$ uptake function. In angiosperms, MCA diverged into two clades and one might have changed functions from proper MCA. However, further studies would be needed to support these hypotheses.

Intriguingly, the E3 ubiquitin ligase type, with MCA^func + U-box + Arm, seems to represent the most ancient MCA^func protein (Fig 1). E3 ubiquitin ligase mediate substrate specificity for ubiquitylation [43] and is a large protein family. In the E3 ubiquitin ligases of *A. thaliana*, only plant U-Box13 (PUB13) and PUB45 retained MCA^func domains (Fig 1; S11 and S16 Appendices). PUB13 was suggested to be involved in the abscisic acid signalling pathway, flowering time, and abiotic stress resistance [44]. The expression level of PUB45 seemed to be affected by nutrients [45]. It could be postulated that the MCA^func domain was first utilized for these E3 ubiquitin ligases for roles for environmental adaptation, though further studies are required here. In addition, there are proteins only retaining the MCA^func domain, but their function is not yet reported and remains unknown (S7 Appendix).

The PLAC8 domain exists in the Plant Cadmium Resistant (PCR) protein family as single domain proteins. PCRs are possibly transmembrane proteins and have roles in cadmium resistance [42] and zinc transport (PCR2; [46]). It is possible that the MCA^func domain, initially part of E3 ubiquitin ligase, and PLAC8, an ion transporter, combined at some point in time and resulted in a novel protein, MCA, as a mechanosensor reacting to environmental calcium ions [11, 14]. The sequences between MCA and other plant mechanosensitive channels, such as MSL, are different [9], and the evolutionary history of MCA is different from that of MSL, which has originated in prokaryotes [47], and may represent an example of convergence in function.

## Conclusions

In conclusion, MCA is an example of a multidomain protein, whose MCA^func domain emerged *de novo* in the ancestor of streptophytes, and recruited an existing domain PLAC8 in the ancestor of embryophytes. The full MCA protein further duplicated and diversified during the evolution of land plants, involving recombination and losses of domains. However, each streptophyte species analysed had at least one complete full MCA copy, pointing to the importance of the protein. The functions of many MCA proteins are not investigated yet but they appear somewhat related to environment sensing, protein-protein interactions, and ion transport. In the basal lineage of streptophytes, *i.e.* charophytes, the MCA^func domain is associated with U-box and Arm domains, supposed to play roles in the E3 ubiquitin ligase pathway. On the other hand, MCA proteins with MCA^func and PLAC8 domains show quite different roles in ion transport. This further supports a hypothesis where domain swapping is an efficient mechanism to increase protein numbers with diversified functions during organismal evolution. Future studies will shed more light on the roles of these proteins and their interactions in relation to land plant evolution.

## Supporting information

**S1 Appendix. Proposed domain structure of MCA.** (*a*) For the MCA protein, the ARPK domain, at the N-terminus, and PLAC8, at the C-terminus, was proposed previously, and it

was shown that MCA has an EF hand-like and a coiled-coil region [7]. (*b*) The biological function of MCA proteins was tested. As a result, the MCA functional domain at the N-terminus, which was previously described as ARPK domain and part of the EF hand-like region, was proposed [18]. This domain was analyzed in the present study and has been registered as MCA^func domain in the Pfam database.
(PDF)

**S2 Appendix. List of proteomes used in this study.**
(PDF)

**S3 Appendix. List of genome and transcriptome databases interrogated in this study, and the list of MCA genes included in the MCA phylogenetic analyses.**
(PDF)

**S4 Appendix. Profile HMM logo of the full MCA protein, PTHR46604:SF3.**
(PDF)

**S5 Appendix. Profile HMM logos of domains in the MCA protein.** (*a*) MCA^func.hmm generated in this study, (*b*) PLAC8.hmm (PF04749).
(PDF)

**S6 Appendix. *MCA* genes found in the genome of *M. polymorpha* subsp. *ruderalis*.** Top: *M. polymorpha* cDNA 0134s0009.1, middle: *M. polymorpha* genome, bottom: *M. polymorpha* subsp. *ruderalis* genome.
(PDF)

**S7 Appendix. MCA^func domain genes retrieved from proteomes.**
(PDF)

**S8 Appendix. PLAC8 domain genes retrieved from proteomes.**
(PDF)

**S9 Appendix. Species tree used for Notung rooting analyses of domain trees.**
(PDF)

**S10 Appendix. Result of the Notung rooting analysis of the MCA^func domain tree.** The possible root position is marked in red (arrow).
(PDF)

**S11 Appendix. MCA^func domain ML tree.** (*a*) Clade associated with E3 ubiquitin ligase AtPUB13. (*b*) Clade associated with AtPUB45. (*c*) MCA^func only proteins. (*d*) Clade associated with AtARO3 and monocot U-box containing protein kinase like proteins. (*e*) MCA clade associated with AtMCA1 and AtMCA2.
(PDF)

**S12 Appendix. Result of the Notung rooting analysis of the PLAC8 domain tree.** Multiple branches show equally strong estimates as possible root positions (marked in red).
(PDF)

**S13 Appendix. PLAC8 domain ML tree.** (*I*) MCA clade (*II*) Clade with proteins retaining DUF2985 + PLAC8.
(PDF)

**S14 Appendix. Notung analyses of the MCA tree.** (*a*) Results of the Notung rooting analyses of the MCA tree. Possible root position is marked in red (arrow). (*b*) Results of Notung rearrangement of the MCA tree. Rearranged branches are marked in yellow. D: inferred

duplication.
(PDF)

**S15 Appendix. Species tree used for Notung analyses for full MCA protein sequence tree.**
(PDF)

**S16 Appendix. Domain partners of MCA<sup>func</sup> domain.** Domain partners observed in the MCA<sup>func</sup> domain containing proteins associated with the tree shown in Fig 1 (left). Domain individual *E* values (i.Evalue) resulting from HMMER website searches are shown as heatmap (right). Absence of domains indicated in grey.
(PDF)

**S17 Appendix. Domain partners of PLAC8 domain.** Domain partners observed in the PLAC8 domain containing proteins associated with the ML tree (left). Domain individual *E* values (i.Evalue) resulting from HMMER website searches are shown as heatmap (right). Absence of domains indicated in grey.
(PDF)

**S18 Appendix. *MCA* gene duplication events estimated by the Notung analysis.** "D" indicates inferred gene duplication events.
(PDF)

**S19 Appendix. The seed alignment and profile HMM of MCA<sup>func</sup> domain.**
(ZIP)

## Acknowledgments

The authors are indebted to Dannie Durand for helpful comments and discussions pertaining to this study, particularly relating to the Notung rooting analyses. The authors also thank Daniel Barker and Frank Wright for helpful discussions. KN is grateful to the following persons for facilitating research associateships, to Pete Hollingsworth and Mark Newman at the Royal Botanic Garden Edinburgh (RBGE), UK, and to Akitoshi Iwamoto at Kanagawa University, Japan. This work was logistically supported by RBGE's Science and ICT divisions. We also thank Duncan Reddish and Catherine Kidner for facilitating RBGE Linux server access and support, and Iain Milne for organizing access to the CropDiversity server, James Hutton Institute, Dundee, UK. We acknowledge the National Institute of Genetics, Japan, for allowing the use of their NIG-supercomputer system. We thank three anonymous reviewers and the editor for their constructive comments.

## Author Contributions

**Conceptualization:** Kanae Nishii, Hidetoshi Iida.

**Data curation:** Kanae Nishii, Michael Möller.

**Formal analysis:** Kanae Nishii, Michael Möller.

**Funding acquisition:** Kanae Nishii, Hidetoshi Iida.

**Investigation:** Kanae Nishii, Michael Möller.

**Methodology:** Kanae Nishii.

**Supervision:** Hidetoshi Iida.

**Writing – original draft:** Kanae Nishii, Michael Möller.

**Writing – review & editing:** Kanae Nishii, Michael Möller, Hidetoshi Iida.

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
