## [Decision Letter · Decision Letter 0]

22 Jan 2021

PONE-D-20-36297

Mix and match: patchwork domain evolution of the land plant-specific Ca2+-permeable mechanosensitive channel MCA

PLOS ONE

Dear Dr. Nishii,

Thank you for submitting your manuscript to PLOS ONE. After careful consideration, we feel that it has merit but does not fully meet PLOS ONE’s publication criteria as it currently stands. Therefore, we invite you to submit a revised version of the manuscript that addresses the points raised during the review process.

This manuscript, and the general appraoch taken, has raised interest in the review team; we encourage you to submit a revision along the lines given further below.

We look forward to receiving your revised manuscript.

Kind regards,

Berthold Heinze

Academic Editor

PLOS ONE

Additional Editor Comments:

The mansucript was reviewed by three independent researchers, who all indicated their (high) interest in the approach and findings. We would therefore very much like to publish an improved version, which takes into account the suggestions of the reviewers. These suggestions are all very clear and to the point, so they should be relatively easy to address (or comment). I am recommending a 'major revision' only in order for the reviewers to have a chance to look at the improved version.

Journal Requirements:

"RBGE is supported by the Rural and Environment Science and Analytical Services Division (RESAS) in the Scottish Government."

"This work was supported by the Japan Society for the Promotion of Science (JSPS) [KAKENHI Grant Number 25120708] to HI, from the Ministry of Education, Culture, Sports, Science & Technology of Japan. KN’s stay at RBGE is financially supported by the Edinburgh Botanic Garden (Sibbald) Trust and the JSPS [JSPS KAKENHI Grant Number 18K06375], and the Sumitomo Foundation."

4. Please upload a new copy of S6 and S17 Figures as the detail is not clear. Please follow the link for more information: https://blogs.plos.org/plos/2019/06/looking-good-tips-for-creating-your-plos-figures-graphics/

Reviewers' comments:

Reviewer's Responses to Questions

**Comments to the Author**

1. Is the manuscript technically sound, and do the data support the conclusions?

Reviewer #1: Partly

Reviewer #2: Yes

Reviewer #3: Yes

2. Has the statistical analysis been performed appropriately and rigorously? 

Reviewer #1: Yes

Reviewer #2: N/A

Reviewer #3: Yes

3. Have the authors made all data underlying the findings in their manuscript fully available?

Reviewer #1: Yes

Reviewer #2: Yes

Reviewer #3: Yes

4. Is the manuscript presented in an intelligible fashion and written in standard English?

Reviewer #1: Yes

Reviewer #2: No

Reviewer #3: Yes

5. Review Comments to the Author

Reviewer #1: In this project the authors explored the evolution of the MCA protein, it’s two domains MCAfunc and PLAC8, and their partner domains across all viridiplantae. They used MCA protein sequences from 55 plant species from various sources in a maximum likelihood phylogenetic analysis of protein domains identified using profile HMMs built using the HMMER program to accomplish this. The authors used their data to explain the origin of MCAfunc, the coupling of MCAfunc and PLAC8, the potential role of duplication in the evolution of the MCA, and briefly to address the role of helper domains in this process. One implication of this study is that MCA may have been fundamental to the evolution of land plants.

I was very impressed with the robustness of the methodology in this work. I cannot imagine a better strategy for building phylogenetic trees or identifying protein domains using this data. I was also impressed with the writing: the abstract is a good summary of the work, the introduction frames the problem well and introduces relevant literature, the results are direct and data-centered yet tell an interesting story, and the discussion does a great job of explaining the results in the context of relevant literature. Although I believe this work is important and well-executed there are some issues with the manuscript itself that the authors need to address, which I have detailed below as either major or minor comments:

Major comment:

1. Major: In many cases the authors refer to a protein or domain evolving in a particular modern group based on their presence in that group as well as in more derived taxa. This is not appropriate without presenting further evidence. The data suggest that the domain or protein most likely arose in the ancestor of all these taxa, which may or may not be a member of the most ancestral group. If the authors know that the common ancestor was a member of the most ancestral modern taxon that evidence must be provided in the manuscript, otherwise the claim must be made only of the common ancestor and not the most ancestral modern taxon. Examples: A) MCAfunc is said to have appeared first in charophytes rather than the ancestor of all streptophytes (lines 29, 303, 348) B) The coupling between MCAfunc and PLAC8 is said to have occurred once, “possibly in the ancestor of bryophytes” (lines 239-242, 348). This should refer to the shared ancestor of all embryophytes. C) several instances in the figure 5 legend. I recommend the authors look for this throughout the paper in case I missed something.

Minor comments:

1. Minor: Some minor grammatical issues were noticed, not enough to be distracting. The authors should edit the full text to identify and resolve as many of these as possible.

2. Minor: The groups “streptophytes” and “chlorophytes” are used extensively here. These terms can be useful, but are sometimes used to mean different things in different manuscripts. Figure 5 provides a visual explanation, but it would be much more helpful if these terms were defined in the introduction to resolve any potential confusion while readers progress through the paper.

3. Minor: Authors use incorrect language when discussing the patterns identified using HMMs referring to them as “HMMs”, when they should be referred to as either “profile HMMs” or “profiles” in brief, not “HMMs”. I double checked this with the HMMER User’s Guide. This issue was noticed on lines 25/26, and 107-114 but may also be present in other places.

4. Minor: I recommend the authors remove the statement about AtMCA1 being expressed in hamster ovary cells on lines 64-66. The contextual difference between plant roots and hamster ovaries are enormous. I therefore found it distracting and I’m not convinced it added to the story, but if the authors disagree this is not something I would insist on.

5. Minor: Authors should expand on the acronym “SMS” on line 134

6. Minor: All figures were low resolution and difficult to read, before publication the authors should replace these with higher resolution versions.

Reviewer #2: During this study, the authors examine the domain evolutionary history of the multi-domain plant-specific mechanosensitive channel, MCA. They thoroughly examined the evolution and origin of both MCA domains MCAfunc and PLAC8, via extensive phylogenetic analysis, and determined that the coupling of both domains occurred via a single event. They reveal that the MCA channel has a complex evolutionary history, however show that at least one full copy of the protein is present in streptophytes, suggesting function of the channel is highly important. The findings presented show an impressive example of how domains can combine, resulting in new protein function and opens up avenues for further studies determining the role of MCA proteins in land plant evolution.

Overall, the study is detailed covering a wide range of plant species and performing extensive analysis to track the evolutionary history of MCA domains. The introduction and discussion sections are clearly written, leading the reader through the logic of the study and are accessible to a wide audience. However, the presentation of the results text is not as clearly written or easy to follow and should be amended prior to publication. As it stands, this section is not accessible to a wider audience.

The results section text lacks suitable explanation and does not provide a narrative leading the reader through the figures. For example the results text starts “In total, 217 proteins possessing…”, without prior explanation (aside from a brief mention in the introduction) as to which species were investigated or the tools by which the search was conducted. In general, this lack of description continues throughout the results text. The authors are recommended to edit the text so that each results section starts by introducing the reasoning behind obtaining the results in the upcoming figure and the brief method by how the results were obtained. The following section should lead from the one before in the text, rather than be stand alone.

Additionally, the presentation of Table 1 as it stands is unclear as the domain contribution types are not mentioned at all until the description of Figure 3. The authors should add a description of these earlier in the text with the presentation of the table, or consider separating this information to a second table which follows Figure 3.

Finally, Appendix S18 is referred significantly within the text and is useful to look at – the authors could consider including this as a main Figure to avoid the reader having to filter through the Supplementary Information.

Additional minor edits/suggestions:

- COMPLEMENTING is spelled incorrectly on line 24 of the abstract

- There is an extremely long sentence between lines 62-68 of the Introduction which should be separated into several sentences.

- In Line 106 of the Methods the authors should define HMM

- Table 1 legend title is vague and is not stand alone- suggest the authors refine this so it is more descriptive

- Suggest adding a column with species in Table 1.

- In line 200, the authors should define ML

- Suggest that the authors add the species to the description of Fig3 in lines 253-266. For example K. nitens (charophyte).

- Appendix S18 is referred to quite a lot within the text and is useful to look at – should this be included as a Figure in its own right to avoid the reader having to filter through the Supplementary Information?

- The resolution of S6 Appendix should be improved if possible- it is difficult to see the residue numbers

- S9 S15 Appendices- only mnemonics are used. Would recommend providing the full names in the legend as well.

- S16 and S17 Appendices- would suggest the authors list the domain partners on the right of the heatmap as in Figures 1 and 2, rather than at the bottom as they are small and hard to see.

Reviewer #3: In the manuscript “Mix and match: patchwork domain evolution of the land plant-specific Ca2+-permeable mechanosensitive channel MCA” the authors present an investigation of the MCA Ca2+ permeable mechanosensitive ion channel using proteomics and bioinformatics. Their data suggests that the MCA protein is composed of many different portions. This work found that the MCA region of a MCA protein is localized to the N-terminus and is ~170 amino acids.

Comments

In lines 188-198 you discuss the Table 1 and mention the different classes of the proteomes, can you include the different classes of the genomes in the table?

You utilize a completed and partial proteome for Marchantia polymorpha as two distinct proteomes, while there is a substantial information to be gained from comparing two different experimental conditions. Are the genes identified in the partial proteome the same as the genes identified in the more complete proteome? Unless there are significant differences between the two proteomes can you eliminate the partial from the data set. If there are substantial differences that are not accounted by the completeness of the proteome then please clarify this in the manuscript.

-As multiple proteomes are incomplete, can you provide the completeness of the proteomes within the manuscript, perhaps in table 1?

-You could remove any incomplete proteomes from the analysis as it is difficult to speculate as if a protein is not present in the data set or not present in the wild type cell.

If the MCA domain is ~420 amino acids (line 71), can you speculate the role of the Type II as they appear to be smaller than the required amount.

In the conclusion can you speculate the role of these proteins and how these are similar or different than the AtMCA1 and AtMCA2 (or other known functional proteins)

In general many of the supplemental figures and figures are blurry and have made some of the interpretation difficult. Many of them required significant magnification in order to see the content. Below you can find some specific suggestions

Figure 1: the clade designations are difficult to see, can you move them to the left on the figure or highlight the regions using specific colors to make it easier to follow?

Figure 3:

As the genes are arranged by species it is unclear if all of the Type Ib are the same, should they be the same?

Is there a difference between the two genes from P. patens?

Are the known MCAs, AtMCA1 and AtMCA2, shown in this graphic?

6. PLOS authors have the option to publish the peer review history of their article (what does this mean?). If published, this will include your full peer review and any attached files.

Reviewer #1: No

Reviewer #2: No

Reviewer #3: No

---

## [Author Response · Author response to Decision Letter 0]

7 Feb 2021

PONE-D-20-36297

Mix and match: patchwork domain evolution of the land plant-specific Ca2+-permeable mechanosensitive channel MCA

Authors` response to the Editor`s general comments

Thank you very much for the supportive comments from the editor. We greatly appreciate the fact that the manuscript has received such a positive interest in our approach and our findings.

We have revised the manuscript following to the editor and reviewer`s comments as detailed below. 

Editor`s comments

"RBGE is supported by the Rural and Environment Science and Analytical Services Division (RESAS) in the Scottish Government."

"This work was supported by the Japan Society for the Promotion of Science (JSPS) [KAKENHI Grant Number 25120708] to HI, from the Ministry of Education, Culture, Sports, Science & Technology of Japan. KN’s stay at RBGE is financially supported by the Edinburgh Botanic Garden (Sibbald) Trust and the JSPS [JSPS KAKENHI Grant Number 18K06375], and the Sumitomo Foundation."

Authors` response

Financial disclosure

"RBGE is supported by the Rural and Environment Science and Analytical Services Division (RESAS) in the Scottish Government."

This sentence is removed from the Acknowledgements and inserted in the Funding Statement section.

The amended Funding Statement is also included in the cover letter.

New Funding Statement is as follows;

"This work was supported by the Japan Society for the Promotion of Science (JSPS) [KAKENHI Grant Number 25120708] to HI, from the Ministry of Education, Culture, Sports, Science & Technology of Japan. KN’s stay at RBGE is financially supported by the Edinburgh Botanic Garden (Sibbald) Trust and the JSPS [JSPS KAKENHI Grant Number 18K06375], and the Sumitomo Foundation. RBGE is supported by the Rural and Environment Science and Analytical Services Division (RESAS) in the Scottish Government."

Editor`s comment

Authors` response

The data submitted to Treebase and protocol.io are now released and available. The Pfam release v.34.0 is under the control of “Pfam” and the “European Bioinformatics Institute” and difficult to arrange individually. Thus we also include the seed alignment of the MCAfunc domain as supplemental data.

“R codes used in this study are published on protocol.io.　[dx.doi.org/10.17504/protocols.io.bkqwkvxe]

Gene trees and their alignment are available from TreeBASE. [http://purl.org/phylo/treebase/phylows/study/TB2:S26880]

The seed alignment of the MCAfunc domain will be available in the next release of the Pfam database. [Pfam release v.34.0; PF19584] as well as the S19 Appendix of this study.”

Editor`s comment

4. Please upload a new copy of S6 and S17 Figures as the detail is not clear.

Authors` Comment

We provided new versions of figures S6 and S17 with higher resolution.

Reviewer 1

Authors` response to the Reviewer 1`s general comments

We like to thank the reviewer for the very positive comments on our work.

Reviewer 1`s comment

1. Major: In many cases the authors refer to a protein or domain evolving in a particular modern group based on their presence in that group as well as in more derived taxa. This is not appropriate without presenting further evidence. The data suggest that the domain or protein most likely arose in the ancestor of all these taxa, which may or may not be a member of the most ancestral group. If the authors know that the common ancestor was a member of the most ancestral modern taxon that evidence must be provided in the manuscript, otherwise the claim must be made only of the common ancestor and not the most ancestral modern taxon. Examples: A) MCAfunc is said to have appeared first in charophytes rather than the ancestor of all streptophytes (lines 29, 303, 348) B) The coupling between MCAfunc and PLAC8 is said to have occurred once, “possibly in the ancestor of bryophytes” (lines 239-242, 348). This should refer to the shared ancestor of all embryophytes. C) several instances in the figure 5 legend. I recommend the authors look for this throughout the paper in case I missed something.

Authors` response

Thank you very much for pointing out this evolutionary issue. We agree that MCAfunc domain might have arisen in the common ancestor of streptophytes, not within charophytes. We also agree that the MCA protein might have originated in the shared ancestor of embryophytes, rather than the ancestor of bryophytes. We have checked the MS and modified all relevant sentences as listed below:

Lines 28-29 [new lines 28-30]

Original text

“Maximum likelihood (ML) analyses revealed that the MCAfunc domain first appeared in E3 ubiquitin ligases-like proteins of charophytes.”

Revised text

“Maximum likelihood (ML) analyses showed that the MCAfunc domain observed in E3 ubiquitin ligases-like proteins of charophytes, suggesting the domain arose in the common ancestor of streptophytes.”

Lines 301-303 [new Lines 332-334]

Original text

“The MCAfunc domain, on the other hand, was streptophyte-specific and not found in chlorophytes, suggesting that the domain originated in charophytes (Fig 5) [4].”

Revised text

“The MCAfunc domain, on the other hand, was streptophyte-specific and not found in chlorophytes, suggesting that the domain originated in the common ancestor of streptophytes, i.e. charophytes plus embryophytes (Fig 6) [4].”

Lines 322-323 [new Lines 354-355]

Original text

“A key event for the MCA evolution seemed to be the partnering of MCAfunc and PLAC8 first recruited in bryophytes.”

Revised text

“A key event for the MCA evolution seemed to be the partnering of MCAfunc and PLAC8 first recruited in the common ancestor of embryophytes.”

Lines 347-348 [new Lines 381-383]

Original text

“In conclusion, MCA is an example of a multidomain protein, whose MCAfunc domain emerged de novo in charophytes, and recruited an existing domain PLAC8 in bryophytes.”

Revised text

“In conclusion, MCA is an example of a multidomain protein, whose MCAfunc domain emerged de novo in the common ancestor of streptophytes, and recruited an existing domain PLAC8 in the common ancestor of embryophytes.”

Lines 239-242 [new Lines 247-250]

Original text

“The ML tree topology and distribution of partner domains suggested that the coupling between MCAfunc and PLAC8 domains occurred once in the plant lineage, possibly in the ancestor of bryophytes, and was sometimes lost after gene duplication events but was always retained in at least one copy.”

Revised text

“The ML tree topology and distribution of partner domains suggested that the coupling between MCAfunc and PLAC8 domains occurred once in the plant lineage, possibly in the common ancestor of embryophytes, and was sometimes lost after gene duplication events but was always retained in at least one copy.”

Line 348

See above

Figure 5 legend

Original text

“Fig 5. Schematic illustration of MCA evolution. PLAC8 domain genes are observed from chlorophytes to angiosperms, and many are single domain proteins (a). In charophytes, type I, type II, and type III domain combinations evolved (b). In the ancestor of land plants, from bryophytes onward, type IV domain combinations (MCA type, i.e. MCAfunc + PLAC8) evolved (c). In addition to the commonly observed types I-IV, lineage specific combinations were also observed.”

Revised text [former Fig. 5 is now Fig. 6, new lines 336-341]

“Fig 6. Schematic illustration of MCA evolution. PLAC8 domain genes were observed from chlorophytes to angiosperms, and many were single domain proteins (a). In the common ancestor of streptophytes, type I, type II, and type III domain combinations evolved (b). In the common ancestor of land plants, i.e. embryophytes, type IV domain combinations (MCA type, i.e. MCAfunc + PLAC8) evolved (c). In addition to the commonly observed types I-IV, lineage specific combinations were also observed.”

Minor comments:

Reviewer 1`s minor comments

1. Minor: Some minor grammatical issues were noticed, not enough to be distracting. The authors should edit the full text to identify and resolve as many of these as possible.

Authors` response

We revised the manuscript and made the appropriate corrections.

Reviewer 1`s minor comments

2. Minor: The groups “streptophytes” and “chlorophytes” are used extensively here. These terms can be useful, but are sometimes used to mean different things in different manuscripts. Figure 5 provides a visual explanation, but it would be much more helpful if these terms were defined in the introduction to resolve any potential confusion while readers progress through the paper.

Authors` response

We added sentences for the definitions of terminology with references.

New Lines 87-91

“Here, for ranks, we followed the definition by Leliaert et al. [19] and NCBI Taxonomy Browser (https://www.ncbi.nlm.nih.gov/guide/taxonomy/), where viridiplantae include green algae (both of chlorophyte and charophyte) and streptophytes, streptophytes include charophyte algae and embryophytes, and embryophytes (also termed as “land plants”) include bryophytes, ferns, gymnosperms and angiosperms.”

Reviewer 1`s minor comments

3. Minor: Authors use incorrect language when discussing the patterns identified using HMMs referring to them as “HMMs”, when they should be referred to as either “profile HMMs” or “profiles” in brief, not “HMMs”. I double checked this with the HMMER User’s Guide. This issue was noticed on lines 25/26, and 107-114 but may also be present in other places.

Authors` response

Thank you very much for pointing out our error. We changed HMMs to profile HMMs throughout the manuscript and legends of figure and appendix.

Reviewer 1`s minor comments

4. Minor: I recommend the authors remove the statement about AtMCA1 being expressed in hamster ovary cells on lines 64-66. The contextual difference between plant roots and hamster ovaries are enormous. I therefore found it distracting and I’m not convinced it added to the story, but if the authors disagree this is not something I would insist on.

Authors` response

The sentence is now removed from lines 64-66

Reviewer 1`s minor comments

5. Minor: Authors should expand on the acronym “SMS” on line 134

Authors` response

The explanation for the acronym SMS has been added in line 134 (new line 139)

“Maximum likelihood (ML) analyses were carried out with PhyML v.3.0 [24] on the ATGC server (www.atgc-montpellier.fr), with Smart Model Selection (SMS) [25].”

Reviewer 1`s minor comments

6. Minor: All figures were low resolution and difficult to read, before publication the authors should replace these with higher resolution versions.

Authors` response

We provided high resolution images for all figures.

General comment from reviewer 2

Reviewer #2: During this study, the authors examine the domain evolutionary history of the multi-domain plant-specific mechanosensitive channel, MCA. They thoroughly examined the evolution and origin of both MCA domains MCAfunc and PLAC8, via extensive phylogenetic analysis, and determined that the coupling of both domains occurred via a single event. They reveal that the MCA channel has a complex evolutionary history, however show that at least one full copy of the protein is present in streptophytes, suggesting function of the channel is highly important. The findings presented show an impressive example of how domains can combine, resulting in new protein function and opens up avenues for further studies determining the role of MCA proteins in land plant evolution.

Overall, the study is detailed covering a wide range of plant species and performing extensive analysis to track the evolutionary history of MCA domains. The introduction and discussion sections are clearly written, leading the reader through the logic of the study and are accessible to a wide audience. However, the presentation of the results text is not as clearly written or easy to follow and should be amended prior to publication. As it stands, this section is not accessible to a wider audience.

Authors` response

Thank you very much for the supportive comment of our work. We revised the manuscript according to the reviewers` and editor`s comments.

Reviewer 2`s comment 1

The results section text lacks suitable explanation and does not provide a narrative leading the reader through the figures. For example, the results text starts “In total, 217 proteins possessing…”, without prior explanation (aside from a brief mention in the introduction) as to which species were investigated or the tools by which the search was conducted. In general, this lack of description continues throughout the results text. The authors are recommended to edit the text so that each results section starts by introducing the reasoning behind obtaining the results in the upcoming figure and the brief method by how the results were obtained. The following section should lead from the one before in the text, rather than be stand alone.

Authors` response

We added narrative sentences at the beginning of each result section.

The species list are in Tables and appendices, and thus we cited those files in the results. Since this study included a large number of species, we like to do this in a table format in Table 1. The details of tools used in this study were stated in the Materials and Method section but we also added short descriptions as follows.

New Lines 181-182

“To determine the distribution of MCA proteins in viridiplantae, 25 proteomes (see Table 1) were interrogated for domains of the MCA protein, MCAfunc and PLAC8, with profile HMMs using HMMER.”

New line 206-207

“Since MCA is a multidomain protein, we studied the phylogenetic relationships of the domains MCAfunc and PLAC8 separately.”

Line 238

“The phylogeny of the other MCA domain, PLAC8, was also examined phylogenetically.”

New lines 261-264

“While the PLAC8 domain commonly existed within the plant, animal, and fungi kingdoms, the MCAfunc domain was only observed in streptophytes in the plant kingdom. Thus, the MCAfunc domain might be the key domain for the MCA protein, and we further assessed the coupling of the MCAfunc domain with its partner domains.”

New lines 291-293

“In order to unravel the history of MCA proteins in plants, a phylogeny of full MCA proteins from 55 embryophyte species was reconstructed. The full MCA proteins include both MCAfunc and PLAC8 domain sequences.”

Reviewer2`s comment 2

Additionally, the presentation of Table 1 as it stands is unclear as the domain contribution types are not mentioned at all until the description of Figure 3. The authors should add a description of these earlier in the text with the presentation of the table, or consider separating this information to a second table which follows Figure 3.

Authors` response

Thank you for this suggestion. The old Table 1 is now split into two, Table 1 and Table 2. Table 2 is placed after Fig.3 for the ease of reading.

Reviewer2`s comment 3

Finally, Appendix S18 is referred significantly within the text and is useful to look at – the authors could consider including this as a main Figure to avoid the reader having to filter through the Supplementary Information.

Authors` response

The S18 Appendix now appears as Fig. 5. The Figure numbers were also amended accordingly.

Reviewer2`s minor comment

Additional minor edits/suggestions:

- COMPLEMENTING is spelled incorrectly on line 24 of the abstract

Authors` response

Thank you for pointing out this error. Now it is corrected.

Reviewer2`s minor comment

- There is an extremely long sentence between lines 62-68 of the Introduction which should be separated into several sentences.

Authors` response

The sentence in lines 62-68 (new lines 63 –70) are revised as follows.

“In A. thaliana, two paralogous MCA genes, AtMCA1 and AtMCA2 have been isolated, and their functions examined in great detail. The AtMCA1 protein is involved in touch sensing at the root tip and a hypoosmotic shock-induced increase in the cytosolic free Ca2+ concentration [7]. AtMCA2 was reported to participate in Ca2+ uptake at the roots [10]. In addition, AtMCA1 and AtMCA2 respond to membrane stretch to generate cation currents when expressed in Xenopus laevis oocytes [8]. Furthermore, MCA channels appear to have common functions in plants, based on studies on Oryza sativa OsMCA1 [11,12,13], Nicotiana tabacum NtMCA1, NtMCA2 [14], Zea mays CNR13 [15], and Streptocarpus MCA-like gene (as Saintpaulia in Ohnishi et al. [16]; see Nishii et al. [17]).”

Reviewer2`s minor comment

- In Line 106 of the Methods the authors should define HMM

Authors` response

The sub-header line 106 (new line 112) has been revised according to the reviewer`s comment and “HMM” is spelled out.

Reviewer2`s minor comment

- Table 1 legend title is vague and is not stand alone- suggest the authors refine this so it is more descriptive

Authors` response

Table 1 is now split to Table 1 and Table 2 following the Reviewer 2`s comment.

The legend of Table 1 has been revised as follows.

“Table 1. Number of proteins found in proteomes. Result of profile HMM searches of MCAfunc and PLAC8 domains in proteomes of 25 taxa across viridiplantae. Number of proteins retaining the MCAfunc or PLAC8 domains (E value < 10-3) are listed and arranged following the Tree of Life (see S7, and S8 Appendix).”

Reviewer2`s minor comment

- Suggest adding a column with species in Table 1.

Authors` response

We are slightly confused by this suggestion, as Table 1 has a “Taxon” column in which the species names are presented.

Reviewer2`s minor comment

- In line 200, the authors should define ML

Authors` response

We added Maximum Likelihood in line 200 (new line 207) following to the reviewer`s comment.

Reviewer2`s minor comment

- Suggest that the authors add the species to the description of Fig3 in lines 253-266. For example K. nitens (charophyte).

Authors` response

In lines 253-266 (new lines 261-279), we added the taxonomic groups to the species.

Reviewer2`s minor comment

- Appendix S18 is referred to quite a lot within the text and is useful to look at – should this be included as a Figure in its own right to avoid the reader having to filter through the Supplementary Information?

Authors` response

Please see above response. S18 Appendix is now as Fig. 5.

Reviewer2`s minor comment

- The resolution of S6 Appendix should be improved if possible- it is difficult to see the residue numbers

Authors` response

The S6 Appendix is now revised and with higher resolution.

Reviewer2`s minor comment

- S9 S15 Appendices- only mnemonics are used. Would recommend providing the full names in the legend as well.

Authors` response

The species` full name was added in the legends of S9 and S15 Appendix. 

Reviewer2`s minor comment

- S16 and S17 Appendices- would suggest the authors list the domain partners on the right of the heatmap as in Figures 1 and 2, rather than at the bottom as they are small and hard to see.

Authors` response

Authors` response

We revised S16 and S17 Appendix following to the reviewer`s comment.

 

Reviewer #3

Reviewer #3: In the manuscript “Mix and match: patchwork domain evolution of the land plant-specific Ca2+-permeable mechanosensitive channel MCA” the authors present an investigation of the MCA Ca2+ permeable mechanosensitive ion channel using proteomics and bioinformatics. Their data suggests that the MCA protein is composed of many different portions. This work found that the MCA region of a MCA protein is localized to the N-terminus and is ~170 amino acids.

Reviewer 3`s comment

In lines 188-198 you discuss the Table 1 and mention the different classes of the proteomes, can you include the different classes of the genomes in the table?

Authors` response

The column “Vernacular name” has been added to Table 1 for the information of taxonomic classes.

Reviewer 3`s comment

You utilize a completed and partial proteome for Marchantia polymorpha as two distinct proteomes, while there is a substantial information to be gained from comparing two different experimental conditions. Are the genes identified in the partial proteome the same as the genes identified in the more complete proteome? Unless there are significant differences between the two proteomes can you eliminate the partial from the data set. If there are substantial differences that are not accounted by the completeness of the proteome then please clarify this in the manuscript.

Authors` response

Although the proteome completeness of Marchantia polymorpha subsp. ruderalis is relatively low, adding this gene to the matrix greatly stabilized the resulting trees. Compared to the well-studied angiosperms, there are only very few proteomes available for the basal groups such as bryophyte or lycophytes, but these are important to add. Therefore, we like to keep this proteome information and sequence in this study.

Reviewer 3`s comment

-As multiple proteomes are incomplete, can you provide the completeness of the proteomes within the manuscript, perhaps in table 1?

Authors` response

The proteome completeness information is added to S2 Appendix. The numbers of proteins registered in each proteomes and their BUSCO completeness were listed. The plaza database of gymnosperm proteomes does not have BUSCO information and thus the information is obtained by the authors. It is added to the materials and methods section.

New lines 107-110

“The proteome completeness information, i.e. BUSCO completeness values (BUSCO-C) were available for most taxa on the Uniprot database. The BUSCO-C values of proteomes from plaza database (Cycas micholitzii, Taxus baccata) were newly obtained in this study using BUSCO v.4.0.6 (Simao et al. 2015), by comparisons against viridiplantae_odb10 lineage datasets.”

Reviewer 3`s comment

-You could remove any incomplete proteomes from the analysis as it is difficult to speculate as if a protein is not present in the data set or not present in the wild type cell.

Authors` response

The proteomes used here represent the reference proteome for the species. Although many proteomes are not complete (most are between 90-99.8%), they are still regarded as the current reference proteomes data for wild type lineages. The phylogenetic placement of the worst complete genome, the gymnosperm Cycas micholitzii (BUSCO-C 36.3%) is consistent with the tree of life and the MCA domain evolution patterns we found. Removal of this and other incomplete proteomes would make the analyses unnecessarily unstable, and we like to keep them included (strictly taken, no proteome is complete, even at BUSCO comparison; see Table S2 Appendix).

Reviewer 3`s comment

If the MCA domain is ~420 amino acids (line 71), can you speculate the role of the Type II as they appear to be smaller than the required amount.

Authors` response

This is a slightly imprecise comment. To be exact, the MCA protein is approximately 420 amino acids in length and retains two domains (MCAfunc and PLAC8). The MCAfunc domain defined in this study is approximately 167 aa length (New line 117).

The Type II domain combination stands for proteins only retaining the MCAfunc domain (see S7 and S16 Appendix). The function of Type II domain genes is unknown in databases such as Uniprot. It would be too speculative to describe function without functional studies, which are much beyond the scope of our study.

We added the following sentence in the discussion section.

New Lines 368-370

“In addition, there are proteins only retaining the MCAfunc domain, but their function is not yet reported and remains unknown (S7 Appendix).”

Reviewer 3`s comment

In the conclusion can you speculate the role of these proteins and how these are similar or different than the AtMCA1 and AtMCA2 (or other known functional proteins)

Authors` response

There is a sentence in the conclusion. We felt that we need to await further studies to be reported to make conclusive statements. We added the following sentences.

New lines 387-392

“In the basal lineage of streptophytes, charophytes, the MCAfunc domain is associated with U-box and Arm domains, supposed to play roles in the E3 ubiquitin ligase pathway. On the other hand, MCA proteins with MCAfunc and PLAC8 domains show quite different roles in ion transport. This further supports a hypothesis where domain swapping is an efficient mechanism to increase protein numbers with diversified functions during organismal evolution.”

Reviewer 3`s comment

In general many of the supplemental figures and figures are blurry and have made some of the interpretation difficult. Many of them required significant magnification in order to see the content. 

Authors` response

We revised all main figures and most supplemental figures, and they now have higher resolution and readable text.

Below you can find some specific suggestions

Reviewer 3`s comment

Figure 1: the clade designations are difficult to see, can you move them to the left on the figure or highlight the regions using specific colors to make it easier to follow?

Authors` response

We added open ellipses to better indicate the clades.

Reviewer 3`s comment

Figure 3:

As the genes are arranged by species it is unclear if all of the Type Ib are the same, should they be the same?

Authors` response

Type Ib share the same domain composition. They sometimes vary in the length of the sequence regions between the domains.

Reviewer 3`s comment

Is there a difference between the two genes from P. patens?

Authors` response

This comment is unclear. In P. patens, three, not two, MCA proteins were recognized. These are slightly different in their amino acid sequences. There are two P. patens specific gene types that vary in their Pkinase and Pkinase_Tyr domains at the 3’end (Fig. 3).

Reviewer 3`s comment

Are the known MCAs, AtMCA1 and AtMCA2, shown in this graphic?

Authors` response

Yes, they are both type IV shown in Fig 3 for A. thaliana, having only the MCAfunc and PLAC8 domains. We added this in Fig. 3 legend.

“The well-studied MCA proteins in A. thaliana, AtMCA1 and AtMCA2, are type IV proteins.”

---

## [Decision Letter · Decision Letter 1]

11 Mar 2021

PONE-D-20-36297R1

Mix and match: patchwork domain evolution of the land plant-specific Ca2+-permeable mechanosensitive channel MCA

PLOS ONE

Dear Dr. Nishii,

Thank you for submitting your manuscript to PLOS ONE. After careful consideration, we feel that it has merit but does not fully meet PLOS ONE’s publication criteria as it currently stands. Therefore, we invite you to submit a revised version of the manuscript that addresses the points raised during the review process.

We request to take on the two remaining issues - a re-worded statement in the Abstract, and better-quality figures (they still appear blurry).

We look forward to receiving your revised manuscript.

Kind regards,

Berthold Heinze

Academic Editor

PLOS ONE

Journal Requirements:

Additional Editor Comments (if provided):

Many thanks for all your work in improving the manuscripts. All reviewers praise the efforts; just two minor issues remain. They can be fixed quite rapidly I hope.

Reviewers' comments:

Reviewer's Responses to Questions

**Comments to the Author**

1. If the authors have adequately addressed your comments raised in a previous round of review and you feel that this manuscript is now acceptable for publication, you may indicate that here to bypass the “Comments to the Author” section, enter your conflict of interest statement in the “Confidential to Editor” section, and submit your "Accept" recommendation.

Reviewer #1: (No Response)

Reviewer #2: All comments have been addressed

Reviewer #3: All comments have been addressed

2. Is the manuscript technically sound, and do the data support the conclusions?

Reviewer #1: Yes

Reviewer #2: Yes

Reviewer #3: (No Response)

3. Has the statistical analysis been performed appropriately and rigorously? 

Reviewer #1: Yes

Reviewer #2: N/A

Reviewer #3: (No Response)

4. Have the authors made all data underlying the findings in their manuscript fully available?

Reviewer #1: Yes

Reviewer #2: Yes

Reviewer #3: (No Response)

5. Is the manuscript presented in an intelligible fashion and written in standard English?

Reviewer #1: Yes

Reviewer #2: Yes

Reviewer #3: (No Response)

6. Review Comments to the Author

Reviewer #1: This manuscript is fundamentally an excellent paper, and it is obvious that the authors have made a strong effort to respond to my comments in the first review response. With two small exceptions I believe it is ready to be accepted for publication at PLOS One. These exceptions are both issues that I mentioned in the first review and will be described in detail below.

1) All figures are still “fuzzy” and difficult to read. I am not sure what the solution is, but I encourage the authors to seek advice and try something new in order to improve their resolution.

2) I identified an issue in my previous review where the evolution of a new domain was attributed to the current most ancestral taxon as opposed to the ancestor of that taxon as well as more derived groups. This was fixed completely in all cases but one. On lines 28-30 in the abstract a claim is still made that the MCAfunc domain first appeared in charophytes, whereas the data suggests it first appeared in the common ancestor of streptophytes. One potential solution would be to change this sentence to, “We identified The MCAfunc domain in all streptophytes including charophytes; based on our Maximum Likelihood (ML) analyses, this suggests the MCAfunc domain evolved early in the history of streptophytes”

Reviewer #2: The authors have suitably addressed all previous comments. Changes to the text and presentations of Tables make for much easier reading.

Reviewer #3: (No Response)

7. PLOS authors have the option to publish the peer review history of their article (what does this mean?). If published, this will include your full peer review and any attached files.

Reviewer #1: No

Reviewer #2: No

Reviewer #3: No

---

## [Author Response · Author response to Decision Letter 1]

16 Mar 2021

PONE-D-20-36297R1

Response to Reviewers

Editor`s comment

We request to take on the two remaining issues - a re-worded statement in the Abstract, and better-quality figures (they still appear blurry).

Reviewer`s comment

Reviewer #1: This manuscript is fundamentally an excellent paper, and it is obvious that the authors have made a strong effort to respond to my comments in the first review response. With two small exceptions I believe it is ready to be accepted for publication at PLOS One. These exceptions are both issues that I mentioned in the first review and will be described in detail below.

1) All figures are still “fuzzy” and difficult to read. I am not sure what the solution is, but I encourage the authors to seek advice and try something new in order to improve their resolution.

2) I identified an issue in my previous review where the evolution of a new domain was attributed to the current most ancestral taxon as opposed to the ancestor of that taxon as well as more derived groups. This was fixed completely in all cases but one. On lines 28-30 in the abstract a claim is still made that the MCAfunc domain first appeared in charophytes, whereas the data suggests it first appeared in the common ancestor of streptophytes. One potential solution would be to change this sentence to, “We identified The MCAfunc domain in all streptophytes including charophytes; based on our Maximum Likelihood (ML) analyses, this suggests the MCAfunc domain evolved early in the history of streptophytes”

Reviewer #2: The authors have suitably addressed all previous comments. Changes to the text and presentations of Tables make for much easier reading.

Reviewer #3: (No Response)

Authors` response

We greatly appreciate the positive comments on our manuscript. Our responses to the specific comments are as follows;

Reviewer 1- Comment 1

We have contacted the PLOS ONE production team regarding the resolution of the figures and they kindly tested our figures’ quality: the figures were tested through PLOS ONE`s imaging tool (PACE) and all figures passed the review. The email from the production teams is attached below. 

We also confirmed that the original TIFF figure files appeared clearly on our screen. We noticed that the generated summary PDF that included all files and figures does not show the figures at the original file quality, and these would not be their final appearance. We are wondering if the reviewer could download (by clicking the box at the top right corner of the figure pages in the pdf file) and review the original TIFF figures, rather than those in the summary PDF.

Reviewer 1- Comment 2

We have revised the sentence along the reviewer`s suggestion.

Lines 28-30

Original sentence:

Maximum likelihood (ML) analyses revealed that the MCAfunc domain first appeared in E3 ubiquitin ligase-like proteins of charophytes, suggesting the domain arose in the common ancestor of streptophytes.

Revised sentences:

We identified the MCAfunc domain in all streptophytes including charophytes where it appeared in E3 ubiquitin ligase-like proteins. Our Maximum Likelihood (ML) analyses suggested that the MCAfunc domain evolved early in the history of streptophytes.

Reviewer 2

We like to thank reviewer 2.

---

## [Editor Report · Decision Letter 2]

24 Mar 2021

Mix and match: patchwork domain evolution of the land plant-specific Ca2+-permeable mechanosensitive channel MCA

PONE-D-20-36297R2

Dear Dr. Nishii,

We’re pleased to inform you that your manuscript has been judged scientifically suitable for publication and will be formally accepted for publication once it meets all outstanding technical requirements.

Kind regards,

Berthold Heinze

Section Editor

PLOS ONE

Additional Editor Comments (optional):

My apologies from the side of the review team for the confusion with figure quality! It is a bit unfortunate that they came out blurry in the process that produced the PDF for review. The originals are acceptable, of course.

So, I am happy to accept this manuscript, which introduces an important new thought - we hope it will receive the attention it deserves.
---

## [Editor Report · Acceptance letter]

5 Apr 2021

PONE-D-20-36297R2 

Mix and match: patchwork domain evolution of the land plant-specific Ca^2+-^permeable mechanosensitive channel MCA 

Dear Dr. Nishii:

I'm pleased to inform you that your manuscript has been deemed suitable for publication in PLOS ONE. Congratulations! Your manuscript is now with our production department. 

Kind regards, 

on behalf of

Dr. Berthold Heinze 

Section Editor

PLOS ONE